**Subject Category:**
Biology (whole organism)

behaviour/cognition/ecology

assessment strategy, contest, game theory, early life, social behaviour, pig

**Author for correspondence:**
Irene Camerlink
e-mail: irene.camerlink@vetmeduni.ac.at

# Advantages of social skills for contest resolution

Irene Camerlink[1,2], Simon P. Turner[1], Marianne Farish[1] and Gareth Arnott[3]

[1]Animal Behaviour and Welfare, Animal and Veterinary Sciences Research Group, Scotland's Rural College (SRUC), West Mains Road, Edinburgh EH9 3JG, UK
[2]Institute of Animal Welfare Science, Department of Farm Animals and Veterinary Public Health, University of Veterinary Medicine Vienna (Vetmeduni), Veterinärplatz 1, 1210 Vienna, Austria
[3]Institute for Global Food Security, School of Biological Sciences, Queen's University, Belfast BT9 7BL, UK

IC, 0000-0002-3427-2210

Animal contests are natural interactions that occur to obtain or defend resources such as food and territory. Selection should favour individuals that can win contests with minimal costs in terms of energy expenditure or injuries. We hypothesized that social skills contribute to animals' assessment abilities in a contest situation and thereby will shorten contest duration. Animals were either raised in early life conditions stimulating the development of social skills, termed socialization or not (control). Contests between 342 pigs at eight weeks old (171 dyads) were studied for opponent assessment ability (using a game theoretical approach), examining duration and escalation, social behaviours performed, injuries and outcome. Contesting dyads were from the same treatment group and varied in body weight, a validated measure of resource holding potential (RHP). Socialized animals had shorter contests that were resolved with fewer injuries and they showed more ritualized display behaviour, consistent with mutual assessment. Furthermore, there was evidence of a novel form of opponent assessment in the socialized group revealed by a positive relationship between winner RHP and fight duration. In conclusion, social skills enabled more rapid establishment of dominance relationships at lower cost. Besides its evolutionary relevance, these findings may also contribute towards improving animal welfare.

## 1. Introduction

Animal contests are ubiquitous and fundamentally important social interactions as they determine the distribution of fitness-relevant resources including territories, mates and food. Selection is expected to favour information gathering that maximizes the benefits and minimizes the costs associated with fighting [1], the

latter including risk of injury and death, energy depletion and increased risk of predation. Contests typically start with displays, following which the encounter may or may not proceed to escalated fighting. Displays may provide information about fighting ability, termed resource holding potential (RHP) [2,3] or aggressive intent [4] (i.e. the intention to attack), offering opponents a mechanism to assess asymmetries and potentially avoid costly fighting.

Being skilful in this assessment process is likely to require experience of social conflict [5]. Gregarious animals typically learn social skills at an early age through playful and agonistic interactions with conspecifics [6,7]. We hypothesize that gregarious animals that lack the possibility during early life to develop social skills appropriate to their species will be less proficient in assessing their opponent in a conflict situation and will consequently show more aggression than socialized animals. This is based on the observation that some domesticated species show high levels of aggression despite such injurious aggression being generally uncommon in their wild ancestors [8]. Although this may in part be related to selective breeding, the environment in which young domesticated animals are reared often lacks opportunities for the development of species-specific social skills.

We studied this hypothesis in pigs (*Sus scrofa*); a species that well illustrates the above example. Wild-living pigs commonly avoid overt aggression [8]. Fights can be costly and injurious but are mostly limited to competition between adult males during the breeding season [9]. In captivity, however, pigs show high amounts of aggression when introduced to an unfamiliar conspecific, even when given the space to retreat [10]. In the wild, piglets first meet non-littermates gradually from 10 days of age [11] but in captivity it is typically abrupt and at four weeks of age. Mimicking this natural situation by co-mingling animals at around two weeks of age, termed early life socialization, has been applied as a method to enhance social skills [12–14]. Exposure to unfamiliar conspecifics in early life typically results in little damaging aggression and provides opportunities for play, including play fighting [15]. Play contributes to the development of cognitive abilities [16,17] and has been hypothesized to influence contest skill and assessment ability [18]. However, to date, it is unknown how early life socialization affects animals' contest skills, and in particular their opponent assessment ability.

Mutual assessment is believed to be the most cognitively advanced form of assessment and is expected to be adopted when the costs of contests are potentially high [19]. In mutual assessment, the individual compares its own RHP to that of the opponent [1,20]. Mutual assessment, as opposed to self-assessment where individuals fight up to their own threshold [21,22], can reduce the contest duration as the weaker individual may retreat when it realizes its inferior position.

Our aim was to investigate whether social skills gained through early life socialization alters the animal's ability to assess an opponent during an agonistic encounter. We hypothesized that social skills make animals capable and/or more proficient in mutual assessment, evident in lower costs in terms of fewer injuries or a shorter contest duration. This was investigated by applying the recommended approach to discriminate between alternative forms of assessment [1], involving the examination of relationships between contestant RHP and various contest costs, including contest duration, fight duration and injuries.

# 2. Material and methods

## 2.1. Animals and housing

A total of 64 sows (Large White × Landrace, served by American Hampshire boars) farrowed in crated pens (3.15 × 1.50 m in total, with a 2.25 × 0.55 m sow crate and 0.65 × 1.50 m heated area for piglets). Piglets were weighed between 6 and 24 h after birth. Males were not castrated and the tail and teeth were kept intact. Every morning pens were cleaned and re-bedded with fresh straw and wood shavings. Piglets received creep feed in a round feeder from approximately 21 days of age and water was available ad libitum. Half of the sows (n = 32) were assigned to a socialization (SOC) treatment, whereas the others (n = 32) were assigned to the control (CON) group. SOC piglets were socialized at 14 days of age by removing the barrier between two adjacent farrowing pens. Socialization of piglets, also termed co-mingling, is the deliberate mingling of two or more litters together pre-weaning to improve piglets' social skills [8,12]. The barrier was then replaced by one of the same design with a 0.36 × 0.75 m opening to allow piglets to move freely between the two pens, which remained until weaning. Only neighbouring sows with less than 48 h between their farrowing were eligible for the socialization treatment. Piglets in the CON treatment remained in their normal farrowing pen with their own sow. From weaning (approx. 26 days of age), pigs (referring to weaned piglets) were then kept in their original sibling group. Pens (1.90 × 5.80 m; approx. 1.1 m²/pig) had a solid floor with

approximately 5 kg of long straw and were cleaned daily. Food and water was provided ad libitum. Pigs were habituated to all test situations to reduce the possibility of fear responses during any of the tests or procedures. This involved gradually exposing pigs (over three occasions) to being alone in a known and unknown area for several minutes and to being handled in a weigh crate.

## 2.2. Contests

Pigs ($n = 364$; eight week age) were matched for paired encounters based on socialization experience, such that dyads consisted of either two socialized or two control pigs. Body weight, a validated measure of RHP in pigs [5], varied between contests in order to include weight matched and unmatched dyads, as recommended to discriminate between assessment models [1]. To achieve this, animals were paired with an opponent of similar (less than 5% difference) or dissimilar body weight (greater than 15% difference). Opponents were genetically unrelated and had not encountered previously. Contests took place in a novel and neutral test arena of $2.9 \times 3.8$ m. The arena had a solid floor with a light bedding of wood shavings and two separate doors to allow opponents to enter the arena simultaneously. The time was started from the moment both had entered the arena fully and was stopped when one pig had retreated and had not shown any aggressive behaviour for 1 min after its retreat (this animal being defined as the loser). Tests were also ended if no winner was apparent after 20 min, or when repeated fear behaviour (escape attempts or loud vocalizations) or mounting behaviour was shown. In the control group, four contests were ended without a clear winner present; two because of a fear response, one because of mounting and one because no winner was apparent within 20 min. In the socialization group, seven contests were ended without a winner present; three because of a fear response, two because of reaching the 20 min threshold and two because of repeated mounting behaviour. These 11 contests (four CON and seven SOC) were excluded from further analysis, giving 171 dyads. The number of skin lesions, i.e. scratches as a result of being bitten, on the whole body were counted for the winner and loser before (pre-contest) and directly after the contest (post-contest) to assess the intensity of fighting. Values pre-contest were subtracted from post-contest values to account for skin lesions already present on the body. Skin lesions are injuries that are here assessed as a form of contest cost.

## 2.3. Behavioural observations

Contests were recorded on video with a Canon Legria HF52 camera which was attached 5 m above the arena. Individuals were recognized by a spray mark (Pig Animal Marker spray) on their back. Contest behaviour was analysed from video by a single observer using The Observer XT 11.5 (Noldus Information Technology). From the video observations, the contest duration was determined (time from entering until leaving the arena) as well as the total fight duration (see table 1 for the definition of a fight). Contests between pigs do not necessarily include a fight. Typically, unfamiliar pigs approach each other and make snout contact, which is then followed by display behaviour such as parallel walking. This can either result in a single bite from one animal, to which the opponent (loser) retreats, or can proceed into a fight in which both opponents mutually bite each other in a rapid sequence until one retreats. Here, contest duration refers to the time from entering the arena until leaving the arena (i.e. when a winner is apparent), whereas fight duration refers only to the exact duration of the opponents being engaged in fighting. All behaviours listed in table 1 were recorded for the latency of their first occurrence within the contest, and their duration in seconds. The duration of each behaviour was expressed as a percentage of the total contest duration. The number of single bites outside fights was recorded as a frequency.

## 2.4. Statistical analyses

Data were analysed with SAS version 9.3 (SAS Institute Inc., Cary, USA). Contest data, including behavioural observations, were analysed by dyad (i.e. pair; $n = 171$ dyads in total). Data regarding body weight were normally distributed; skin lesion data (count data) were skewed and were square root (sqrt) transformed to reach normality; durations were log transformed and proportions of time spent on a behaviour were arcsine sqrt transformed to reach normality. Results are presented as means with standard error or as back-transformed values when indicated.

Assessment strategies were analysed by the relationship between contest costs (contest duration, fight duration and the number of skin lesions) and winner and loser RHP for SOC and CON dyads. Body weight was used as a proxy-measure for RHP. Contest duration, being the duration from entering the arena until a winner is apparent, was included as a traditional measure of contest cost. Fight duration,

**Table 1.** Ethogram. Reduced ethogram showing the main behaviours for display, non-damaging aggression (NDA) and damaging aggression (DA). For the full ethogram, see electronic supplementary material, table S1.

| behaviours | category | description |
|---|---|---|
| investigation[a] | | sniff or light touch of body of other pig with the nose |
| heads up | display | both pigs have their nose/head lifted parallel or frontal towards each other |
| parallel walking | display | pigs walk or trot simultaneously with the shoulders next to each other |
| shoulder-to-shoulder | display | standing or moving with the shoulder against the shoulder of the other whereby heads are frontal without real pressure on the shoulder |
| orientation[a] | | pig stands still, walks or repositions itself while orienting towards the opponent in between agonistic interactions |
| pushing | NDA | pig uses its head or shoulder to move the opponent while putting pressure on the shoulder |
| nose-wrestling | NDA | pig firmly presses the side of its nose against the side of the nose of the opponent |
| single bite[a] | DA | pig delivers a bite which contacts and injures the other pig; each single bite was recorded when it happened outside of a fight |
| fight | DA | pig delivers an aggressive act which the opponent retaliates to with an aggressive act within 5 s |
| bullying | DA | one pig pursues the other, chasing and biting or attempting to bite as the other withdraws |
| retreat[a] | | pig abruptly turns its head away from the opponent and does not show any aggressive behaviour within 10 s |
| non-agonistic[a] | other | all non-agonistic behaviours such as walking, standing, lying down, investigating the area |

[a]Behaviours are not mutual but instead recorded separately for each individual.

being the exact duration of fighting behaviour, was taken as a more accurate measure of contest costs [5] and was only evaluated for contests that included a fight (133 out of 171 contests; 77%). The number of skin lesions, for winners and losers separately, was also included as it is currently the best proxy-measure for contest costs in pigs [5]. These contest costs were analysed as the dependent variables in separate mixed models (Proc MIXED) with as fixed effect the interactions winner RHP × SOC/CON treatment and loser RHP × SOC/CON treatment to analyse the relationships between contest costs and winner and loser RHP, a validated approach to discriminate between assessment models for different treatment groups [5]. Other fixed effects in the model were the main effect of SOC/CON treatment, the weight difference between the opponents and opponents' sex (MM/MF/FF). Variables were omitted from the model only if their removal improved the model fit as assessed through the AIC and BIC. Batch (i.e. farrowing group) was included as random effect.

Fight occurrence (yes/no) was analysed using a generalized linear mixed model (Proc GLIMMIX) with a binary distribution and logit link function. Fight occurrence was the response variable, with SOC/CON treatment, weight difference and opponents' sex as fixed effects and batch as a random variable.

Data on the duration/proportion of time spent on mutual behaviours, the latency until the first occurrence of each behaviour, and the number (frequency) of single bites (summed by dyad) were analysed by dyad in separate mixed models (Proc MIXED) for each of the behaviours (table 1). The behaviour was the response variable and the fixed effects were the SOC/CON treatment, the weight difference between the opponents, and opponents' sex (MM/MF/FF). Batch was included as random effect.

## 3. Results

Winners were slightly heavier than losers (winners 22.43 ± 0.199 kg; losers 21.67 ± 0.188; $t_{143} = 2.38$; $p = 0.019$), but in the contests with a weight difference between the opponents the chance of the heaviest opponent winning was not significantly greater than that the chance of the lightest opponent winning ($\chi^2 = 1.993$; $p = 0.18$). In 55.9% of the contests, the heaviest opponent won but in 44.1% of the contests

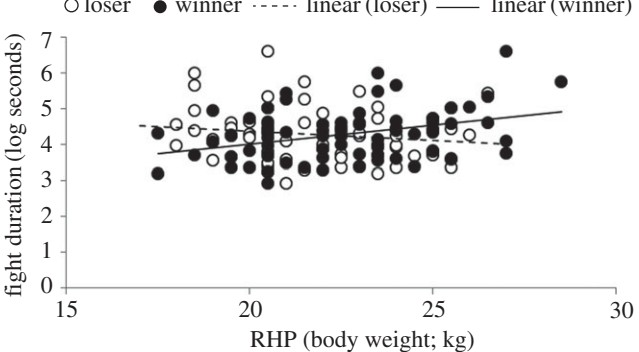

**Figure 1.** Relationship between fight duration and winner and loser RHP for the socialized group.

**Table 2.** Means with s.e. for the percentage of the total contest duration spent on mutual behaviours, for socialized ($n = 93$) and control dyads ($n = 79$).

| % of time | socialized | control | F-statistics | p-value |
|---|---|---|---|---|
| nose wrestling | 3.6 ± 0.42 | 3.7 ± 0.48 | 1,79 = 0.01 | 0.93 |
| shoulder-to-shoulder | 9.9 ± 0.81 | 9.6 ± 1.07 | 1,148 = 0.97 | 0.33 |
| parallel walking | 8.1 ± 0.84 | 5.7 ± 0.52 | 1,128 = 3.73 | 0.06 |
| heads up | 4.2 ± 0.42 | 4.2 ± 0.51 | 1,132 = 0.27 | 0.60 |
| pushing | 8.0 ± 1.26 | 8.4 ± 1.22 | 1,97 = 0.19 | 0.67 |
| fighting | 18.6 ± 1.73 | 21.1 ± 1.89 | 1,122 = 0.72 | 0.40 |
| bullying | 13.0 ± 1.45 | 13.8 ± 1.78 | 1,135 = 0.09 | 0.76 |
| non-agonistic | 37.0 ± 1.70 | 32.1 ± 2.01 | 1,161 = 3.31 | 0.07 |

the lightest opponent won. This situation did not significantly differ between SOC and CON dyads ($\chi^2 = 0.064$; $p = 0.86$). When the smallest opponent won, it was on average $2.79 \pm 0.289$ kg (12.7%) lighter than its opponent.

Early life socialization of pigs decreased their contest duration at eight weeks of age as compared to non-socialized animals ($F_{1,161} = 5.21$; $p = 0.02$). The average contest duration of SOC dyads was 187 s (CI 162–212), whereas for CON dyads this was 230 s (CI 198–262; back-transformed values). Winner and loser RHP did not influence contest duration within SOC or CON dyads (all $p > 0.05$).

Differences in the fight duration were assessed over 133 contests as 39 (23%) did not escalate into a fight. The percentage of contests without a fight did not differ between the treatment groups (SOC 26%; CON 19%; $F_{1,161} = 1.47$; $p = 0.23$). There was no difference in fight duration between SOC and CON ($F_{1,122} = 0.80$; $p = 0.37$). The interaction between RHP and SOC/CON treatment was significant for winners ($F_{2,118} = 4.18$; $p = 0.02$) but not for losers ($F_{2,118} = 1.04$; $p = 0.36$). Within SOC dyads, the fight duration increased with winner RHP ($b = 0.108 \pm 0.038$; $F_{2,118} = 2.86$; $p = 0.005$), but it was unaffected by loser RHP ($b = -0.010 \pm 0.044$; $F_{2,118} = -0.24$; $p = 0.81$; figure 1). In CON dyads, the fight duration was unaffected by winner RHP ($b = -0.013 \pm 0.049$; $F_{2,118} = -0.26$; $p = 0.80$) or loser RHP ($b = 0.067 \pm 0.047$; $F_{2,118} = 1.42$; $p = 0.16$).

Costs in terms of injuries were lower in SOC dyads for winners ($F_{1,161} = 4.90$; $p = 0.03$) and losers ($F_{1,161} = 7.86$; $p = 0.006$) as compared to CON. In CON dyads, winners had on average $32.3 \pm 4.70$ skin lesions and they inflicted on average $65.8 \pm 6.64$ lesions on the losers. In SOC dyads, winners had only $20.8 \pm 2.88$ lesions and inflicted on average $45.2 \pm 4.09$ lesions on the losers. As the fight duration did not differ between the treatments, the difference in skin lesions can be ascribed to the single bites in the time outside fights (described below) or due to the rate of biting during the fight. The number of skin lesions, both on the winner's and loser's body, was unaffected by winner and loser RHP, for CON and SOC dyads (all $p > 0.05$).

The proportion of time spent on each behaviour was largely similar between the treatment groups, meaning that the contest dynamics were similar despite the shorter contest duration in SOC dyads (table 2). Socialized pigs tended to show relatively more parallel walking, a form of display

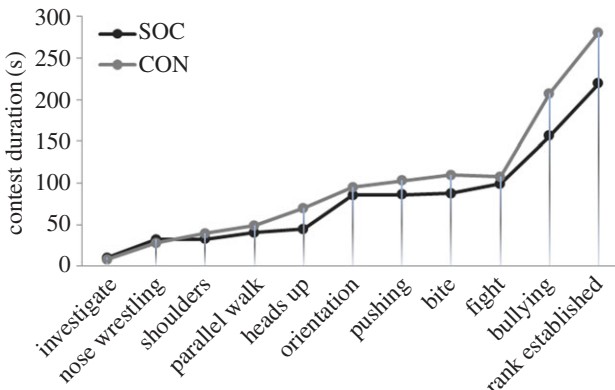

**Figure 2.** Latencies until the average display of the specified agonistic behaviours, showing contest escalation over time for the socialized (SOC) and control group (CON).

behaviour, and more non-agonistic behaviour (walking, standing, exploring the environment) than control pigs (table 2). Within SOC dyads, animals tended to bite each other less frequently outside of a fight than animals in CON dyads (SOC $3.9 \pm 0.19$ sqrt (15 bites); CON $4.4 \pm 0.20$ sqrt (20 bites); $F_{1,152} = 3.37$; $p = 0.07$).

The latency until a behaviour was shown did not differ between the socialized and control groups (figure 2; $p > 0.10$). The shorter contest duration was therefore due to the accumulation of measurable but non-significantly shorter durations spent on the various behaviours. The shorter contest duration may also be due to differences in the frequency of single behaviours, typically lasting not more than 1–2 s, which were counted as point events rather than durations (e.g. bites and head knocks). The proportion of time spent on behaviours and the latency until they began was not significantly influenced by the difference in body weight between the opponents, varying from 0 to 32% weight difference, or by the opponents' sex.

## 4. Discussion

Early life socialization resulted in changes in later life agonistic behaviour compared to animals deprived of this opportunity. Socialized animals had shorter contests that were resolved with fewer injuries (skin lesions), with a trend for less biting, and increased use of a display behaviour termed parallel walking that has been hypothesized to assist opponent assessment in other species (e.g. *Dama dama* [23,24]). In summary, early life socialization enabled the more rapid establishment of dominance relationships at lower cost.

This is consistent with an enhanced ability to estimate relative differences in RHP, i.e. mutual assessment. However, using the recommended statistical approach of examining relationships between winner and loser RHP and contest costs to discriminate between alternative forms of assessment [1], revealed no conclusive support for any particular strategy, an issue that has been documented for a range of other contest studies [25–27]. In the socialized group, there was a positive relationship between winner RHP and fight duration. This is opposite to the prediction for mutual assessment whereby the relationship is negative [1,28]. However, this assumes that fight motivation decreases with the detection of increased opponent ability, i.e. the bigger the winner, the more quickly the loser perceives this and retreats. The current result of a positive relationship is consistent with increased fight motivation by the smaller contestant as winner ability increases and is therefore still consistent with a form of mutual assessment, albeit in the opposite direction to initial predictions. Indeed, a number of contest models illustrate increased aggression in small individuals facing larger opponents [29–32]. For example, the Napoleon strategy illustrates that if RHP is not completely predictive of fight outcome (i.e. that sometimes the smaller individual wins), then individuals can be as aggressive as their larger opponents [32]. Here, in 44% of the contests the opponent with the smallest RHP (lowest body weight) beat an opponent that was heavier. On average, they overcame a body weight difference of 12.7% (2.79 kg). The 'desperado effect' [29] illustrates that small individuals may attack because they have few alternative opportunities to obtain resources. Thus, animal contest researchers should be open to the possibility that fight motivation can increase as well as decrease with increasing

opponent fighting ability. When put together with the differences in contest behaviour, the current results are consistent with early life socialization equipping pigs with enhanced social skills for resolving agonistic encounters, likely mediated through an improved assessment ability.

Consistent with our findings, early life experiences including play and play fighting have been hypothesized as important for the development of contest skill [18]. To explore this further, we are currently pursuing research examining how play behaviour differs between socialized and non-socialized animals, and how cognitive ability is influenced in later life. This latter point is important given cognitive ability for animal contest behaviour, although hypothesized to be important, has been largely overlooked to date [33].

Studies involving early life socialization of piglets show differences in the frequency and/or the duration of biting behaviour at post-weaning regrouping [8,12]. While D'Eath & Turner [8] reported that socialized pigs were quicker to attack a small, unfamiliar intruder introduced into the home pen, others (e.g. [12]) found that socialized pigs took longer to attack a new pig. This discrepancy in attack latency is likely to result from the different social contexts in which aggressiveness was tested. Both scenarios did, however, result in less aggression overall in socialized animals. Social contact between unfamiliar animals in early life therefore seems to allow the more rapid acquisition of mature social skills and/or cognitive ability. Encouraging social skills in early life can benefit social species in captivity, such as dogs [34] and zoo animals including primates [6]. Aggressive behaviour in captive species is of major animal welfare concern [10,34]. Encouraging the development of social skills in early life therefore contributes directly to animal welfare, with the reduction in injuries due to fighting being one of a range of benefits offered by socialization.

Ethics. This study was approved by SRUC's animal experiments committee and was carried out under UK Home Office licence (project licence PPL60/4330), and in constant collaboration with SRUC's named veterinary surgeon. The study was carried out in accordance with the recommendation in the European Guidelines for accommodation and care of animals, UK Government DEFRA animal welfare codes, and adhered to the ASAB/ABS guidelines. Strict endpoints were in place for the termination of contests to ensure that any harm to the welfare of the animals was kept at a minimum. This prevented any injury other than skin lesions due to receiving bites.
Data accessibility. Primary data are provided in the electronic supplementary material.
Authors' contributions. All authors contributed in the design of the experiment; I.C. and M.F. carried out the animal work; I.C. carried out the statistical analyses; I.C. and G.A. wrote the manuscript; M.F. and S.P.T. helped draft the manuscript. All authors gave final approval for publication.
Competing interests. We declare we have no competing interests.
Funding. This research was funded by the Biotechnology and Biological Sciences Research Council (BBSRC). SRUC receives financial support from the Scottish Government.
Acknowledgements. We thank Agnieszka Futro and Mhairi Jack for their help with carrying out the experimental work.

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
