## [Reviewer comments · Royal Society Open Science]

Review History

RSOS-181456.R0 (Original submission)

Review form: Reviewer 1

Is the manuscript scientifically sound in its present form?

Yes

Are the interpretations and conclusions justified by the results?

Yes

Is the language acceptable?

Yes

Is it clear how to access all supporting data?

Yes

Do you have any ethical concerns with this paper?

No

Have you any concerns about statistical analyses in this paper?

Yes

Recommendation?

Accept with minor revision (please list in comments)

Comments to the Author(s)

Review note on RSOS-181456 "Advantages of social skills for contest resolution"

General comments

This is a well written and well-structured manuscript. The study has a good design, is clearly presented and the conclusions reflects the results. I have some minor specific comments and some questions related to the statistical analyses (see below).

Specific comments

Abstract

LN 22: It would help the reader if you add the age of the pigs at the contest.

LN 28: Write out what RHP stands for, this is the first time the reader meets this abbreviation.

Materials and methods

LN 110-111: Please specify which fear behaviors you refer to.

LN 121-129: Please clarify how the individuals were identified on the films (i.e. how they were marked).

LN 131 and onwards: Statistical analyses:

- The treatments (SOC and CON) were allocated to sows (specified in line 84 and 85), thus the analyses should be done on averages per litter (sow) and not on individual piglet level (as sow/litter is the experimental unit).
- Specify which effects in the model are fixed and which are random.
- Even though the correct way to analyses this data is on litter/sow level and not piglet level. If analyzed on piglet level (given that this is well justified in the statistical analyses section), the random effect of litter(or sow if each sow only have one litter in the experiment) should be included as a random effect in the model.

Results

Throughout the results section: Give one more decimal for SE than for the least square mean, now you give the same number of decimals for both LSM and SE.

Review form: Reviewer 2**Is the manuscript scientifically sound in its present form?**

Yes

Are the interpretations and conclusions justified by the results?

Yes

Is the language acceptable?

Yes

Is it clear how to access all supporting data?

Yes

Do you have any ethical concerns with this paper?

No

Have you any concerns about statistical analyses in this paper?

No

Recommendation?

Accept with minor revision (please list in comments)

Comments to the Author(s)

In this study, the authors test the effects of early life socialisation on contest dynamics and intensity in pigs. They hypothesise that socialised pigs will be less aggressive than non-socialised pigs due to an increased ability to assess their opponent. The authors find that socialised pigs are indeed less aggressive during fights, being able to resolve conflict in a less costly manner than non-socialised pigs. However, they find mixed evidence as to whether or not this is due to an increased assessment ability.

Overall this is a neat study with interesting results. I have only minor comments for the authors to consider before publication can be recommended.

INTRODUCTION

I found the structure of the introduction a little unusual, specifically with the hypothesis of the study being presented in the second paragraph! However, all the information is there to set up the study so unless the editor has a problem with this I think it's OK albeit unconventional.

Line 42: Please define 'aggressive intent'.

Line 75 and others: I think you need to be specific when you talk about 'contest costs'. Traditionally when examining assessment rules, one would inspect the relationship between contestant RHP and contest duration (as a proxy of cost) but you also explore injury here (which is very cool). I think it would be worthwhile for you to be explicit throughout the manuscript when talking about costs to make it clear to the reader specifically which proxy of cost you are referring to.

METHODS

Line 8: Why was the barrier replaced if the piglets were still able to mingle? Was this to prevent aggression between the sows?

Line 91 and others: Be more specific about which animals. Say piglets or pigs depending on which you are referring to.

Line 100: Are pigs still classed as piglets at 8 weeks old? Please specify.

Line 117: Were pre-fight lesions the result of play-fighting between siblings or were they only present on SOC piglets? Did you look at the effect the presence of these pre-fight lesions had on contest behaviour? Could be quite interesting to see if pigs with these lesions were more or less aggressive regardless of socialisation treatment?

Line 125-126: Table 1 only gives a definition of 'mutual fighting' not 'fighting'. I find the use of fight to mean something different to contest very confusing throughout the manuscript and recommend that you are explicit about the difference between these two terms when you first being to use them. I also recommend that you are consistent with using 'mutual fight' rather than

just 'fight' to avoid confusion. It also might be worth writing a brief description of how a pig fight plays out in this section so the reader understands all that comes next.

Line 138: Again be specific about what you mean when you say 'contest costs' here.

Line 150: What is 'batch'?

RESULTS

Lines 191-192: "... shorter contest duration was therefore due to the accumulation of non-significant time differences" - I don't understand this sentence.

Review form: Reviewer 3

Is the manuscript scientifically sound in its present form?

Yes

Are the interpretations and conclusions justified by the results?

Yes

Is the language acceptable?

Yes

Is it clear how to access all supporting data?

Yes

Do you have any ethical concerns with this paper?

No

Have you any concerns about statistical analyses in this paper?

No

Recommendation?

Accept with minor revision (please list in comments)

Comments to the Author(s)

This study looked at whether socialization influences the aggressive behavior of pigs. The study design is straightforward and the authors bring in tests from contest theory to see not only whether overall levels of aggression differed but also whether the way that individuals assess one another may depend on socialization. And indeed it seems that socialization affects both things, and importantly reduces contest duration and costs, which has potentially very important welfare implications. I found the study design and write-up very well done, and I only have a few minor comments related to the interpretation of the data. It's a bit difficult to tell what's going on here in terms of assessment, which the authors acknowledge, but I think the possibilities are even broader than what they discuss.

Specific comments:

Line 72: Just to be clear here, is the hypothesis that socialized pigs should show mutual assessment and other pigs should do something else (self assessment?)? Or rather that both types should do mutual assessment, but socialized pigs do it better.

Line 185: Although Table 1 is cited above when the definition of fights is brought up, it would probably help to alert the reader to the fact that all these other behaviors were also being measured, sometime in the Methods section. This result on parallel walking comes as a surprise otherwise because we wouldn't have known that was looked at.

Line 191: Which "shorter contest duration" is being referred to here? Same comment for next sentence.

Line 200: They were shorter contests, but presumably the fight is the most costly part of the contest? If they're both fighting equally often and for equal amounts of time, what does this say about differences in aggression and assessment in these species? (granted, the trend for less biting does of course suggest lower costs for socialized pigs)

Line 214: Would this strategy be based on absolute winner size, or winner size relative to loser size? In the latter case, then wouldn't you also expect a negative relationship between loser size and contest duration? Did the larger pig always win?

Line 225: The results on overall contest duration are consistent with pigs being better at resolving encounters, but it's not clear if that's because of improved assessment. Isn't one interpretation of the results that the individuals were actually doing a worse job of assessing one another, assuming mutual assessment? That is, shouldn't large socialized winners, if they were good mutual assessors, been especially good at defeating their opponent, and shouldn't small socialized losers have been very good at giving up right away when facing a larger opponent? Instead, large socialized winners seemed to take an especially long time to win for some reason, which didn't happen in the control group.

Figure 1: Y axis is log scale, but still not clear what the original units of time were.

Decision letter (RSOS-181456.R0)

24-Apr-2019

Dear Dr Camerlink

On behalf of the Editors, I am pleased to inform you that your Manuscript RSOS-181456 entitled "Advantages of social skills for contest resolution" has been accepted for publication in Royal Society Open Science subject to minor revision in accordance with the referee suggestions. Please find the referees' comments at the end of this email.

The reviewers and handling editors have recommended publication, but also suggest some minor revisions to your manuscript. Therefore, I invite you to respond to the comments and revise your manuscript.

- Ethics statement

- Data accessibility

If you wish to submit your supporting data or code to Dryad (<http://datadryad.org/>), or modify your current submission to dryad, please use the following link:
<http://datadryad.org/submit?journalID=RSOS&manu=RSOS-181456>

- Competing interests

- Authors' contributions

- Acknowledgements

- Funding statement

Because the schedule for publication is very tight, it is a condition of publication that you submit the revised version of your manuscript before 03-May-2019. Please note that the revision

deadline will expire at 00.00am on this date. If you do not think you will be able to meet this date please let me know immediately.

If your manuscript is newly submitted and subsequently accepted for publication, you will be asked to pay the article processing charge, unless you request a waiver and this is approved by

Royal Society Publishing. You can find out more about the charges at <http://rsos.royalsocietypublishing.org/page/charges>. Should you have any queries, please contact openscience@royalsociety.org.

Kind regards,
Andrew Dunn
Royal Society Open Science
openscience@royalsociety.org

on behalf of Dr Safi Darden (Associate Editor) and Kevin Padian (Subject Editor)
openscience@royalsociety.org

Reviewer comments to Author:
Reviewer: 1

Comments to the Author(s)
Review note on RSOS-181456 "Advantages of social skills for contest resolution"

General comments

This is a well written and well-structured manuscript. The study has a good design, is clearly presented and the conclusions reflects the results. I have some minor specific comments and some questions related to the statistical analyses (see below).

Specific comments

Abstract

LN 22: It would help the reader if you add the age of the pigs at the contest.

LN 28: Write out what RHP stands for, this is the first time the reader meets this abbreviation.

Materials and methods

LN 110-111: Please specify which fear behaviors you refer to.

LN 121-129: Please clarify how the individuals were identified on the films (i.e. how they were marked).

LN 131 and onwards: Statistical analyses:

- The treatments (SOC and CON) were allocated to sows (specified in line 84 and 85), thus the analyses should be done on averages per litter (sow) and not on individual piglet level (as sow/litter is the experimental unit).
- Specify which effects in the model are fixed and which are random.
- Even though the correct way to analyses this data is on litter/sow level and not piglet level. If analyzed on piglet level (given that this is well justified in the statistical analyses section), the random effect of litter(or sow if each sow only have one litter in the experiment) should be included as a random effect in the model.

Results

Throughout the results section: Give one more decimal for SE than for the least square mean, now you give the same number of decimals for both LSM and SE.

Reviewer: 2

Comments to the Author(s)

In this study, the authors test the effects of early life socialisation on contest dynamics and intensity in pigs. They hypothesise that socialised pigs will be less aggressive than non-socialised pigs due to an increased ability to assess their opponent. The authors find that socialised pigs are indeed less aggressive during fights, being able to resolve conflict in a less costly manner than non-socialised pigs. However, they find mixed evidence as to whether or not this is due to an increased assessment ability.

Overall this is a neat study with interesting results. I have only minor comments for the authors to consider before publication can be recommended.

INTRODUCTION

I found the structure of the introduction a little unusual, specifically with the hypothesis of the study being presented in the second paragraph! However, all the information is there to set up the study so unless the editor has a problem with this I think it's OK albeit unconventional.

Line 42: Please define 'aggressive intent'.

Line 75 and others: I think you need to be specific when you talk about 'contest costs'. Traditionally when examining assessment rules, one would inspect the relationship between contestant RHP and contest duration (as a proxy of cost) but you also explore injury here (which is very cool). I think it would be worthwhile for you to be explicit throughout the manuscript when talking about costs to make it clear to the reader specifically which proxy of cost you are referring to.

METHODS

Line 8: Why was the barrier replaced if the piglets were still able to mingle? Was this to prevent aggression between the sows?

Line 91 and others: Be more specific about which animals. Say piglets or pigs depending on which you are referring to.

Line 100: Are pigs still classed as piglets at 8 weeks old? Please specify.

Line 117: Were pre-fight lesions the result of play-fighting between siblings or were they only present on SOC piglets? Did you look at the effect the presence of these pre-fight lesions had on contest behaviour? Could be quite interesting to see if pigs with these lesions were more or less aggressive regardless of socialisation treatment?

Line 125-126: Table 1 only gives a definition of 'mutual fighting' not 'fighting'. I find the use of fight to mean something different to contest very confusing throughout the manuscript and recommend that you are explicit about the difference between these two terms when you first being to use them. I also recommend that you are consistent with using 'mutual fight' rather than just 'fight' to avoid confusion. It also might be worth writing a brief description of how a pig fight plays out in this section so the reader understands all that comes next.

Line 138: Again be specific about what you mean when you say 'contest costs' here.

Line 150: What is 'batch'?

RESULTS

Lines 191-192: "... shorter contest duration was therefore due to the accumulation of non-significant time differences" – I don't understand this sentence.

Reviewer: 3

Comments to the Author(s)

This study looked at whether socialization influences the aggressive behavior of pigs. The study design is straightforward and the authors bring in tests from contest theory to see not only whether overall levels of aggression differed but also whether the way that individuals assess one another may depend on socialization. And indeed it seems that socialization affects both things, and importantly reduces contest duration and costs, which has potentially very important welfare implications. I found the study design and write-up very well done, and I only have a few minor comments related to the interpretation of the data. It's a bit difficult to tell what's going on here in terms of assessment, which the authors acknowledge, but I think the possibilities are even broader than what they discuss.

Specific comments:

Line 72: Just to be clear here, is the hypothesis that socialized pigs should show mutual assessment and other pigs should do something else (self assessment?)? Or rather that both types should do mutual assessment, but socialized pigs do it better.

Line 185: Although Table 1 is cited above when the definition of fights is brought up, it would probably help to alert the reader to the fact that all these other behaviors were also being measured, sometime in the Methods section. This result on parallel walking comes as a surprise otherwise because we wouldn't have known that was looked at.

Line 191: Which "shorter contest duration" is being referred to here? Same comment for next sentence.

Line 200: They were shorter contests, but presumably the fight is the most costly part of the contest? If they're both fighting equally often and for equal amounts of time, what does this say about differences in aggression and assessment in these species? (granted, the trend for less biting does of course suggest lower costs for socialized pigs)

Line 214: Would this strategy be based on absolute winner size, or winner size relative to loser size? In the latter case, then wouldn't you also expect a negative relationship between loser size and contest duration? Did the larger pig always win?

Line 225: The results on overall contest duration are consistent with pigs being better at resolving encounters, but it's not clear if that's because of improved assessment. Isn't one interpretation of the results that the individuals were actually doing a worse job of assessing one another, assuming mutual assessment? That is, shouldn't large socialized winners, if they were good mutual assessors, been especially good at defeating their opponent, and shouldn't small socialized losers have been very good at giving up right away when facing a larger opponent? Instead, large socialized winners seemed to take an especially long time to win for some reason, which didn't happen in the control group.

Figure 1: Y axis is log scale, but still not clear what the original units of time were.

Author's Response to Decision Letter for (RSOS-181456.R0)

See Appendix A.

Decision letter (RSOS-181456.R1)

03-May-2019

Dear Dr Camerlink,

I am pleased to inform you that your manuscript entitled "Advantages of social skills for contest resolution" is now accepted for publication in Royal Society Open Science.

on behalf of Dr Safi Darden (Associate Editor) and Kevin Padian (Subject Editor)
openscience@royalsociety.org

Follow Royal Society Publishing on Twitter: [@RSocPublishing](https://twitter.com/RSocPublishing)
Follow Royal Society Publishing on Facebook:
<https://www.facebook.com/RoyalSocietyPublishing.FanPage/>
Read Royal Society Publishing's blog: <https://blogs.royalsociety.org/publishing/>

Appendix A

Response to Referees

Manuscript ID RSOS-181456 "Advantages of social skills for contest resolution"

Editor:

On behalf of the Editors, I am pleased to inform you that your Manuscript RSOS-181456 entitled "Advantages of social skills for contest resolution" has been accepted for publication in Royal Society Open Science subject to minor revision in accordance with the referee suggestions. Please find the referees' comments at the end of this email.

Reviewer: 1

This is a well written and well-structured manuscript. The study has a good design, is clearly presented and the conclusions reflects the results. I have some minor specific comments and some questions related to the statistical analyses (see below).

Specific comments

Abstract

LN 22: It would help the reader if you add the age of the pigs at the contest.

Response: done, age added

LN 28: Write out what RHP stands for, this is the first time the reader meets this abbreviation.

Response: done

Materials and methods

LN 110-111: Please specify which fear behaviors you refer to.

Response: done, escape attempts or loud vocalizations

LN 121-129: Please clarify how the individuals were identified on the films (i.e. how they were marked).

Response: done, individuals were recognized by a spray mark (Pig Animal Marker spray) on their back

LN 131 and onwards: Statistical analyses:

- The treatments (SOC and CON) were allocated to sows (specified in line 84 and 85), thus the analyses should be done on averages per litter (sow) and not on individual piglet level (as sow/litter is the experimental unit).

Response: all measures were taken on the dyad level, with the dyad being the experimental unit.

- Specify which effects in the model are fixed and which are random.

Response: done

- Even though the correct way to analyses this data is on litter/sow level and not piglet level. If analyzed on piglet level (given that this is well justified in the statistical analyses section), the random effect of litter(or sow if each sow only have one litter in the experiment) should be included as a random effect in the model.

Response: because data was analysed on dyad level (not individual pig level) the sow/litter cannot be included as random effect because a dyad consists of two non-littermates. In previous

work where the data was analysed on individual level (different experimental set-up) the sow/litter had no influence on contest outcomes.

Results

Throughout the results section: Give one more decimal for SE than for the least square mean, now you give the same number of decimals for both LSM and SE.

Response: done

Reviewer: 2

In this study, the authors test the effects of early life socialisation on contest dynamics and intensity in pigs. They hypothesise that socialised pigs will be less aggressive than non-socialised pigs due to an increased ability to assess their opponent. The authors find that socialised pigs are indeed less aggressive during fights, being able to resolve conflict in a less costly manner than non-socialised pigs. However, they find mixed evidence as to whether or not this is due to an increased assessment ability.

Overall this is a neat study with interesting results. I have only minor comments for the authors to consider before publication can be recommended.

INTRODUCTION

I found the structure of the introduction a little unusual, specifically with the hypothesis of the study being presented in the second paragraph! However, all the information is there to set up the study so unless the editor has a problem with this I think it's OK albeit unconventional.

Line 42: Please define 'aggressive intent'.

Response: done, i.e. the intention to attack.

Line 75 and others: I think you need to be specific when you talk about 'contest costs'. Traditionally when examining assessment rules, one would inspect the relationship between contestant RHP and contest duration (as a proxy of cost) but you also explore injury here (which is very cool). I think it would be worthwhile for you to be explicit throughout the manuscript when talking about costs to make it clear to the reader specifically which proxy of cost you are referring to.

Response: done, throughout the manuscript contest cost has been specified when it was about a specific measure. Also in line 121 added: 'Skin lesions are injuries that are here assessed as a form of contest cost.' Using injury as contest cost is not new (e.g. Rudin & Briffa, 2011, Proc Roy Soc B) but indeed not yet traditional.

METHODS

Line 8: Why was the barrier replaced if the piglets were still able to mingle? Was this to prevent aggression between the sows?

Response: indeed, although sows could not physically contact each other even without barrier, it was placed to reduce stress in the sows from being in close eye contact to another. More details on this and an image of the set-up were published in Camerlink et al. 2019 Animals.

Line 91 and others: Be more specific about which animals. Say piglets or pigs depending on which you are referring to.

Response: piglet refers to before weaning whereas pig refers to weaned animals. This has been specified now in line 92 and 'animals' replaced by the type of pig.

Line 100: Are pigs still classed as piglets at 8 weeks old? Please specify.

Response: no, clarified in line with above remark.

Line 117: Were pre-fight lesions the result of play-fighting between siblings or were they only present on SOC piglets? Did you look at the effect the presence of these pre-fight lesions had on contest behaviour? Could be quite interesting to see if pigs with these lesions were more or less aggressive regardless of socialisation treatment?

Response: pre-contest lesions are present on both SOC and CON pigs (average CON 16.5 ± 1.06 ; SOC 14.4 ± 0.88) and are more likely a result of minor aggression (single bites) than play fighting. The frequency of play fighting in these pigs pre-weaning did not relate to skin lesions at 2 weeks of age. Pre-test lesions are usually only used as correction for the number of fresh lesions. Here they weakly correlated with lesions post contest ($r = 0.29$).

Line 125-126: Table 1 only gives a definition of 'mutual fighting' not 'fighting'. I find the use of fight to mean something different to contest very confusing throughout the manuscript and recommend that you are explicit about the difference between these two terms when you first being to use them. I also recommend that you are consistent with using 'mutual fight' rather than just 'fight' to avoid confusion. It also might be worth writing a brief description of how a pig fight plays out in this section so the reader understands all that comes next.

Response: done, description added (line 129-133 new manuscript) and throughout the manuscript 'mutual fight' has changed into 'fight' for ease of understanding (a fight is always mutual in our definition).

Line 138: Again be specific about what you mean when you say 'contest costs' here.

Response: done

Line 150: What is 'batch'?

Response: farrowing group (added to the text).

RESULTS

Lines 191-192: "... shorter contest duration was therefore due to the accumulation of non-significant time differences" – I don't understand this sentence.

Response: revised to 'The shorter contest duration was therefore due to the accumulation of numerical but non-significant shorter durations spent on the various behaviours.'

Reviewer: 3

This study looked at whether socialization influences the aggressive behavior of pigs. The study design is straightforward and the authors bring in tests from contest theory to see not only whether overall levels of aggression differed but also whether the way that individuals assess one another may depend on socialization. And indeed it seems that socialization affects both things, and importantly reduces contest duration and costs, which has potentially very important welfare implications. I found the study design and write-up very well done, and I only have a few minor comments related to the interpretation of the data. It's a bit difficult to tell what's going on here in terms of assessment, which the authors acknowledge, but I think the possibilities are even broader than what they discuss.

Specific comments:

Line 72: Just to be clear here, is the hypothesis that socialized pigs should show mutual assessment and other pigs should do something else (self assessment?)? Or rather that both types should do mutual assessment, but socialized pigs do it better.

Response: both, sentence reworded. Pigs generally do not show mutual assessment (except for a few), unless they gained fighting experience. Socialization (whereby they fight at early age) was hypothesized to increase the number of pigs showing mutual assessment as well as the proficiency herein.

Line 185: Although Table 1 is cited above when the definition of fights is brought up, it would probably help to alert the reader to the fact that all these other behaviors were also being measured, sometime in the Methods section. This result on parallel walking comes as a surprise otherwise because we wouldn't have known that was looked at.

Response: done. Line 129-133 in revised manuscript.

Line 191: Which "shorter contest duration" is being referred to here? Same comment for next sentence.

Response: contest duration refers to the whole contest. The following has been added to the methods for clarification: 'Here, contest duration refers to the time from entering the arena until leaving the arena (when a winner is apparent) whereas fight duration refers only to the exact duration of the opponents being engaged in fighting.'

Line 200: They were shorter contests, but presumably the fight is the most costly part of the contest? If they're both fighting equally often and for equal amounts of time, what does this say about differences in aggression and assessment in these species? (granted, the trend for less biting does of course suggest lower costs for socialized pigs)

Response: skin lesions reflect the in intensity of fighting. Thus although the fight duration did not differ, the number of skin lesions indicates that the fights were less intense. For assessment the display phase is more important, as they can then still 'decide' to not fight at all.

Line 214: Would this strategy be based on absolute winner size, or winner size relative to loser size? In the latter case, then wouldn't you also expect a negative relationship between loser size and contest duration? Did the larger pig always win?

Response: here the models are based on absolute size of the winner and loser. In pigs, the largest opponent is not always the winner but on average the winner is usually larger. In fact, in 37% of contests the smaller opponent won and only in 47% the largest won (in 15% the winner and loser had the same weight). This data has now been added in line 174-178 (1st paragraph Results) of the revised manuscript and been mentioned in the Discussion.

Line 225: The results on overall contest duration are consistent with pigs being better at resolving encounters, but it's not clear if that's because of improved assessment. Isn't one interpretation of the results that the individuals were actually doing a worse job of assessing one another, assuming mutual assessment? That is, shouldn't large socialized winners, if they were good mutual assessors, been especially good at defeating their opponent, and shouldn't small socialized losers have been very good at giving up right away when facing a larger opponent? Instead, large socialized winners seemed to take an especially long time to win for some reason, which didn't happen in the control group.

Response: This points out exactly why injuries (skin lesions) are (at least for pigs) a better measure of contest costs than the traditional measure of duration. For pigs it is more advantageous to spent more time in display behaviour (non-damaging assessment of the situation) to avoid costly and injurious fighting. Good assessors may therefore have a longer contest duration but should have fewer injuries.

Figure 1: Y axis is log scale, but still not clear what the original units of time were.

Response: seconds, axis adjusted.